# *Aerococcus viridans* Phage Lysin AVPL Had Lytic Activity against *Streptococcus suis* in a Mouse Bacteremia Model

**DOI:** 10.3390/ijms242316670

**Published:** 2023-11-23

**Authors:** Hengyu Xi, Yao Fu, Chong Chen, Xin Feng, Wenyu Han, Jingmin Gu, Yalu Ji

**Affiliations:** 1State Key Laboratory for Diagnosis and Treatment of Severe Zoonotic Infectious Diseases, Key Laboratory for Zoonosis Research of the Ministry of Education, Institute of Zoonosis, College of Veterinary Medicine, Jilin University, Changchun 130062, China; m15543135579@163.com (H.X.); fuyao0519@126.com (Y.F.); m13944830520@163.com (C.C.); hanwy@jlu.edu.cn (W.H.); jingmin0629@163.com (J.G.); 2Jiangsu Co-Innovation Center for the Prevention and Control of Important Animal Infectious Diseases and Zoonoses, Yangzhou University, Yangzhou 225009, China

**Keywords:** *Streptococcus suis*, *Aerococcus viridans*, biofilm, bacteremia, phage lysin

## Abstract

*Streptococcus suis* (*S. suis*) is a swine pathogen that can cause sepsis, meningitis, endocarditis, and other infectious diseases; it is also a zoonotic pathogen that has caused a global surge in fatal human infections. The widespread prevalence of multidrug-resistant *S. suis* strains and the decline in novel antibiotic candidates have necessitated the development of alternative antimicrobial agents. In this study, AVPL, the *Aerococcus viridans* (*A. viridans*) phage lysin, was found to exhibit efficient bactericidal activity and broad lytic activity against multiple serotypes of *S. suis*. A final concentration of 300 μg/mL AVPL reduced *S. suis* counts by 4–4.5 log10 within 1 h in vitro. Importantly, AVPL effectively inhibited 48 h *S. suis* biofilm formation and disrupted preformed biofilms. In a mouse model, 300 μg/mouse AVPL protected 100% of mice from infection following the administration of lethal doses of multidrug-resistant *S. suis* type 2 (SS2) strain SC19, reduced the bacterial load in different organs, and effectively alleviated inflammation and histopathological damage in infected mice. These data suggest that AVPL is a valuable candidate antimicrobial agent for treating *S. suis* infections.

## 1. Introduction

*Streptococcus suis* is a Gram-positive pathogen widely distributed throughout the world that has caused three fatal outbreaks in humans as an emerging zoonotic pathogen [1,2]. The main route of *S. suis* infection in pigs is upper respiratory tract infection; meanwhile, people are infected mainly through direct or indirect contact with sick pigs or highly contaminated meat products [3]. Humans and pigs often show clinical symptoms, such as arthritis, pneumonia, and endocarditis, after infection, which can lead to meningitis, bacteremia, and even death [3]. To date, *S. suis* has been mainly classified into 29 serotypes (1–19, 21, 23–25, 27–31, and 1/2), of which SS2 has been reported as the most virulent type in humans and pigs [4,5]. Although *S. suis* is considered a reservoir of antibiotic resistance, as it has played a role in the horizontal transfer of resistance genes [6,7], antibiotics are still the major tool for controlling *S. suis* infections, which pose a serious threat to animal husbandry and human health. In addition, *S. suis* can aggregate to form biofilms that play a role in its pathogenesis, making antibiotic therapy more challenging [8]. Consequently, the development of nonantibiotic agents against *S. suis* is important.

Lysin, a peptidoglycan hydrolase derived from phages, is increasingly drawing attention as a candidate novel antimicrobial agent to combat infections caused by Gram-positive bacteria. Lysin has the advantages of high specificity and efficiency, as well as low levels of drug resistance [9]. As a potential antibacterial agent, lysin has shown great potential in various biomedical applications, and its effectiveness has been verified in various animal models of bacterial infections [10,11,12], with some lysins already in human clinical trials [13]. However, the high degree of specificity of phage lysin is both an advantage and a limitation. Clinical pathogenic infections may be caused by multiple bacteria simultaneously, in which case, a broad bactericidal spectrum is considered a prerequisite for a therapeutic candidate [14,15]. Therefore, phage lysins with broad lytic activity should be developed for the treatment of infections caused by different pathogenic bacteria.

In our previous study, AVPL (GenBank accession number: AXY81212) was hypothesized to be a lysin from the *A. viridans* phage vB_AviM_AVP based on genome analysis [16], with 48% coverage and 59.17% identity to N-acetylmuramoyl-L-alanine amidase. In this work, we found that AVPL has cross-generic bactericidal activity against different serotypes of *S. suis*. The effects of AVPL on *S. suis* biofilm prevention and elimination were evaluated in vitro. In addition, our therapeutic results with lysin AVPL in a mouse model of bacteremia provide a potential strategy for the treatment of *S. suis*-induced infections.

## 2. Results

### 2.1. The Lytic Activity of AVPL against Different Serotypes of S. suis

The purified and soluble lysin AVPL showed the right mass and near homogeneity (approximately 55 kDa), as revealed by SDS-PAGE (Figure 1A). In addition, through structural analysis of lysin AVPL, which contains an amidase catalytic domain and two LysM binding domains, the fusion proteins of each of the truncated domains with EGFP are shown in Figure 1A, as EGFP-AVPL-C, EGFP-AVPL-B1, EGFP-AVPL-B2, and EGFP-AVPL-B1B2, with sizes of approximately 43, 39, 50, and 53 kDa, respectively. The spot-testing experiment revealed that AVPL showed effective bactericidal activity against *S. suis*. However, there was no inhibitory activity against other Gram-positive bacteria, such as *Streptococcus pneumoniae* (*S. pneumoniae*), *Staphylococcus aureus* (*S. aureus*), *Enterococcus faecalis* (*E. faecalis*), and *Enterococcus faecium* (*E. faecium*), or Gram-negative organisms (Appendix A).

AVPL was found to be highly active against all *S. suis* strains used in this study, including *S. suis* serotypes 2, 3, 7, 9, 12, and 27 (Figure 1B). Although variations in the lysis of different strains were observed, the OD_600_ value of 15/30 *S. suis* strains decreased by approximately 0.5 after 30 min of AVPL treatment.

Time- and dose-dependent catalytic activity indicated that the effective bactericidal activity of AVPL against SS2 and SS9 strains fully occurred within 60 min, resulting in a ~2 log10 reduction in the number of viable cells with 50 μg/mL AVPL treatment (Figure 1C). The killing activity increased with increasing enzyme concentration, and 300 μg/mL AVPL resulted in a 4–4.5 log10 reduction.

The susceptibility of 30 *S. suis* strains was determined against eight antibiotics in this study (Appendix A), and strains resistant to three or more antibiotics were considered multidrug resistant [17]. The results showed that the majority (22/30) of *S. suis* strains tested were multidrug-resistant bacteria, with high resistance rates of 93.33% and 83.33% to tetracycline and lincomycin, respectively. A high proportion of strains were also resistant to ceftriaxone (14/30) and erythromycin (17/30), but all strains were susceptible to meropenem.

### 2.2. Binding Activity of the Truncated Fragments of AVPL to S. suis

The lytic activity of AVPL targets *S. suis*, so we hypothesized that AVPL should maintain the ability to bind to *S. suis*. As shown in Figure 2, EGFP-AVPL-B1-, EGFP-AVPL-B2-, and EGFP-AVPL-B1B2- *S. suis* cells all exhibited green EGFP fluorescence, indicating that these proteins can bind to the surface of *S. suis* cells. In contrast, *S. suis* treated with EGFP-AVPL-C, which lacks the binding domain, did not emit green fluorescence.

### 2.3. Antibiofilm Activity of AVPL

Of all 23 strains of *S. suis* tested, 4 showed the ability to form strong biofilms and 9 formed moderate biofilms (Appendix A). SC493, with strong biofilm formation ability, was used for subsequent experiments. As shown in Figure 3A–D, the semiquantitative determination of biofilm formation via crystal violet staining showed that 20 μg/mL AVPL treatment effectively inhibited 48 h biofilm formation and disrupted the mature biofilm when compared to that of the control group. Additionally, AVPL also showed a moderate effect on the 72 h biofilm.

The effect of AVPL in combating biofilms was also illustrated in SEM micrographs (Figure 3E). *S. suis* formed a luxuriant biofilm with a thick growth of chain-like bacteria in the biofilm. However, the biofilm pretreated with 20 µg/mL AVPL was thinner than the control biofilm, bacterial cells were dispersed and aggregated less, and almost no intact chains were observed. Furthermore, compared to the dense biofilm of the control group, most of the 48 h biofilm was disrupted after 1 h of exposure to AVPL, and the peptidoglycan layer was hydrolyzed or even disintegrated, leading to a mass of cellular debris in the pictures. Consequently, AVPL prevents biofilm formation and promotes the degradation of preformed biofilms.

### 2.4. AVPL Protects Mice against Lethal S. suis Infection

Assessment of the in vivo efficacy of AVPL was conducted in a systemic *S. suis* infection model. A single intraperitoneal injection of 7 × 10^7^ CFU/mouse of SC19 failed to cause total mortality, whereas 7 × 10^8^ CFU/mouse or 7 × 10^9^ CFU/mouse of SC19 was sufficient to cause a 100% mortality rate within 24 h (Appendix A). One hour after the infection of mice with a lethal dose (7 × 10^8^ CFU/mouse) of SC19, a considerable amount of bacteria (~1.75 × 10^6^ CFU/mL) was detectable in peripheral blood, indicating rapid spread from the abdominal cavity, leading to systemic infection (Appendix A). Therefore, 1 h after challenge was considered the appropriate time for AVPL administration. As shown in Figure 4A, AVPL exhibited a dose-dependent protective effect, with 300 μg/mouse of AVPL sufficient to provide 100% protection from death, while 7-day survival rates for mice receiving 50 μg/mouse or 150 μg/mouse were 10% and 70%, respectively. Within 24 h of *S. suis* infection, 100% (*n* = 10) of the control mice died from bacteremia. In addition, a single high dose of AVPL (600 μg/mouse) did not elicit significant changes in health or survival during the 7-day follow-up period (Appendix A). The safety of AVPL was further confirmed by pathological histological analysis, as shown in Figure 5A. Following a single high-dose AVPL treatment, the tissues were morphologically intact and without inflammatory cell infiltration, similar to that of normal tissues.

The in vivo therapeutic potential of AVPL was further evaluated by comparing bacterial loads in the blood and organs (lung, liver, spleen, and kidney) of mice in different treatment groups postinfection. After intraperitoneal injection of 7 × 10^8^ CFU/mouse of SC19, the bacterial load in the blood of mice in the phosphate buffer treatment group rapidly reached >10^6^ CFU/mL within 1 h and continued to increase to 10^9^ CFU/mL over time, culminating in death within 24 h (Figure 4B). In contrast, AVPL treatment resulted in a steady decline in the blood bacterial load over time to 4.9 × 10^3^ at 48 h. Figure 4C–F show that the bacterial load in the lung and liver reached approximately 10^6^ CFU/mg at 6 h postinfection in the control group and remained without significant change until 24 h. The bacterial load of the spleen and kidney decreased slightly after reaching a peak at 6 h. In contrast, AVPL treatment at 1 h postinfection effectively reduced the bacterial load in each organ to <10^3^ CFU/mg at 24 h.

Further pathological histological analysis of the organs was performed. Compared to normal mice, 24 h after infection with *S. suis* SC19, mice in the phosphate buffer-treated group showed a severe inflammatory response in the lung tissue, with thickened alveolar septa, dilated capillaries, and neutrophil infiltration (Figure 5A). There was neutrophil infiltration in the liver tissue with some cell necrosis, as well as bruising and thrombosis. Bruising and lymphocytic infiltration were also observed in the spleen tissue. In contrast, AVPL showed significantly reduced pathological damage in every tissue after 24 h of treatment, and pathology at 48 h was not significantly different from that of the normal group.

The levels of cytokines in the blood of mice receiving different treatments were measured. As shown in Figure 5B, serum TNF-α and IL-6 levels in the phosphate buffer-treated group peaked at 6 h (3.8 × 10^3^ pg/mL) and 12 h (4.8 × 10^3^ pg/mL), respectively, and then declined gradually. However, TNF-α levels in the AVPL-treated group were significantly lower after 6 h and IL-6 levels after 12 h compared to the phosphate buffer-treated group, but the levels were similar to the normal group at 24 h.

## 3. Discussion

*S. suis* has caused considerable economic losses in the pig industry for decades. *S. suis* asymptomatically colonizes approximately 80% of pigs; however, it has emerged with increasing frequency as a pathogen of human infection since the first human case of *S. suis* was reported in Denmark [18,19]. Meanwhile, drug resistance in *S. suis* caused by conventional antibiotic therapy has become a serious issue, with an increasing number of resistant strains, including those resistant to tetracyclines, macrolides-lincomycins-streptogramin B (MLS_B_), phenolics, and oxazolidinones [1,6,20,21]. Among these antibiotics, oxazolidinones are considered the “last line of defense” against Gram-positive bacteria [22]. Importantly, mobile genetic elements make *S. suis* a reservoir of antimicrobial resistance and underlie its contribution to the widespread dissemination of resistance genes among bacteria [1]. Of the 30 strains of *S. suis* isolated from clinical cases in this study, 73.33% were multidrug-resistant, indicating that clinically effective control of *S. suis* infections is becoming a major challenge. Therefore, new technologies and strategies are urgently needed in the treatment, prevention, and control of drug-resistant bacterial infections.

Phage lysins are a new class of antibacterial drugs used to prevent and treat bacterial infections [23]. Since only five *Streptococcus* virulent phages have been identified (and only one *S. suis* virulent phage) [24], the reported streptococcal lysins are mainly derived from prophages carried by different strains, a limited number of which are effective against *S. suis* [10,25,26,27,28,29,30,31]. Therefore, the exploration of phage lysins that exhibit high activity against *S. suis* has been challenging. Additionally, previously reported lysins generally show a narrow lytic spectrum of activity against one species or certain isolates of one species [10,32]. In general, the development of chimeric lysin is a promising approach to broadening the host spectrum [33]. In this study, the *A. viridans* phage lysin AVPL exhibited a broad range of lytic activity against two different genera of Gram-positive pathogens (*A. viridans* and *S. suis*). This cross-generic lytic activity has rarely been reported with other lysins. AVPL is the first and only *A. viridans* phage lysin reported thus far, and it is unknown whether this feature is common to all *A. viridans* phage lysins or is specific to AVPL. More importantly, AVPL efficiently lysed all *S. suis* strains used in the study, including those of serotypes 2, 3, 7, 9, 12, and 27. The wide bactericidal spectrum indicates that this lysin has potential for application [34].

The high diversity in peptidoglycan structure and glycan chains among different Gram-positive bacteria results in the lytic activity of lysins rarely being effective in different species or even being limited to certain serotypes [32]. In contrast, streptococcal lysins, such as PlySs2, PlyC, and PlySK1249, all have broad activity against multiple groups of streptococci [10,26,35]. The antimicrobial activity of lysins against several streptococcal species is mainly due to a common architecture in streptococcal peptidoglycan, especially the cross-bridges [36]. However, the cross-generic lytic activity of AVPL was only against *S. suis* and was ineffective against the tested *S. equi*, *S. dysgalactiae*, *S. agalactiae*, or *S. pneumoniae* strains. Both *A. viridans* and *S. suis* belong to the Lactobacillales order, so there may be similarities in the peptidoglycan structures, but there are no reports of streptococcal phage lysins capable of lysing *A. viridans*. Laser scanning confocal microscopy (LSCM) observations also indicated that the binding spectrum of AVPL is consistent with its activity range, which is in line with the properties of most phage lysins, different from lysin PlySs2, which lyses a wide range of *Streptococcus*, *Staphylococcus*, and *Listeria* strains but whose binding activity is limited to *Streptococcus* [37]. The binding region is known to determine the substrate-binding site and, therefore, the activity the lysin [33]. Future studies related to protein structure and interaction with substrates could be used to more precisely elucidate the molecular mechanism of the cross-generic lytic activity of the lysin AVPL.

Various serotype strains of *S. suis* were recently found to form biofilms [38,39], which play a key role in the development of certain pathologies, such as meningitis and endocarditis [40,41,42]. Notably, the formation of biofilms may lead to the failure of almost all antibiotic treatments, including antibiotics from different classes, making the disease’s eradication and recurrence difficult [8,43]. *S. suis* has been demonstrated to cause persistent infection through biofilm formation, posing a problem for disease prevention and control [44]. Thus far, only *S. suis* phage lysin LySMP and chimeric lysin Csl2 have been reported to eradicate the biofilm of *S. suis* [45,46]. In this study, 20 μg/mL AVPL effectively inhibited and eradicated 48 h *S. suis* biofilms. The inhibition of biofilm formation by AVPL is most likely due to its rapid bactericidal activity, resulting in a reduction in the number of bacteria available for biofilm formation [47]. The biofilm matrix includes a variety of components, such as extracellular polysaccharides, proteins, and DNA, and although lysins do not act directly on these matrix constituents, the bacteria inside the biofilm are still the main component of biofilms; thus, AVPL-mediated lysis of *S. suis* weakened an important structural component of the biofilm and contributed to the elimination of biofilms [48,49]. Both the staphylococcal phage-derived lysin P128 and the *S. suis* lysin LySMP have been reported to disrupt biofilms to a greater extent when used in combination with antibiotics [45,50]. This may be related to the lysin-mediated increase in bacterial cell wall permeability, thus allowing for improved antibiotic activity. Considering the above factors, combining AVPL with antibiotics that target bacteria or depolymerases that specialize in the degradation of extracellular polysaccharides may provide better biofilm clearance and increase bactericidal activity, thus leading to more effective treatment of biofilm-associated infections [51].

In the bacteremia model, a single dose of AVPL was effective in reducing the *S. suis* load in the blood and major organs of infected mice, as well as the levels of inflammatory factors. Moreover, residual bacteria randomly collected from the blood and different organs of mice at the late stage of treatment remained susceptible to lysin AVPL. Meanwhile, most streptococcal infections occur on exposed surfaces or externally accessible mucosal surfaces, which is beneficial for therapy with phage lysin AVPL [36]. These results suggest that the *A. viridans* phage-derived lysin AVPL has great potential in controlling multidrug-resistant *S. suis* infections while the effects of lysin on immune cells and in vivo metabolism need to be further investigated.

*A. viridans* is a Gram-positive coccus belonging to the genus *Aerococcus* of the family *Aerococcaceae* and is a rarely studied zoonotic pathogen. It is morphologically similar to streptococci and can also clinically cause bacteremia, endocarditis, urinary tract infections, meningitis, and arthritis in humans and pigs [52,53]. In addition, co-infection with *A. viridans* and *S. suis* has recently been reported to cause meningitis in pigs and mice [54]. Considering the frequent coexistence of these two bacteria in multiple clinical diseases, since AVPL has shown good bactericidal activity against both bacteria in vitro and in vivo, it should be able to effectively treat co-infectious diseases caused by *A. viridans* and *S. suis*.

## 4. Materials and Methods

### 4.1. Bacterial Strain Culture

All *Streptococcus*, *S. aureus*, *E. faecalis*, and *E. faecium* strains were routinely cultured using brain heart infusion (BHI) broth (Becton, Dickinson and Company, Franklin Lakes, NJ, USA) with shaking (180 rpm) at 37 °C. *Salmonella*, *Klebsiella pneumoniae*, and *Acinetobacter baumannii* were cultured in Luria–Bertani (LB) broth. The solid medium contained 1.5% agar. All strains were stored at −80 °C in 30% (*v*/*v*) glycerol.

*S. suis* was incubated on Mueller–Hinton agar (MHA) plates supplemented with 5% (*v*/*v*) defibrinated sheep blood at 37 °C. Antimicrobial susceptibility testing was performed using the paper diffusion method according to Clinical and Laboratory Standard Institute (CLSI) guideline M100-ED31. Breakpoints of *Viridans. streptococci* were used in this experiment as there are no breakpoints available for *S. suis*.

### 4.2. Constructs, Protein Expression, and Purification

The primers utilized in this study are listed in Appendix A. The gene for AVPL was amplified by polymerase chain reaction and cloned into the pMCSG7 vector, as previously reported [55]. The fusion genes EGFP-AVPL-C (EGFP-catalytic domain), EGFP-AVPL-B1 (EGFP-binding domain 1), EGFP-AVPL-B2 (EGFP-binding domain 2), and EGFP-AVPL-B1B2 (EGFP-binding domain 1 + 2) were constructed with pET-15b vector and the Seamless Cloning Kit (Shanghai Beyotime Biotechnology Co., Ltd., Shanghai, China) according to the manufacturer’s instructions. After sequencing, the correct plasmid was transformed into *E. coli* BL21 (DE3). Proteins were expressed, purified, and prepared according to previously described methods [56].

### 4.3. Bactericidal Assessment of AVPL

Details of the strains involved in this study are shown in Appendix A. A spot test was first used to determine the lytic activity of AVPL against a variety of Gram-positive and Gram-negative bacteria.

Strains that could be lysed were further assayed via the turbidity reduction assay for lysis activity. Different serotypes of *S. suis* cultured to an OD_600_ of 0.6–0.8 were washed three times with sterile phosphate buffer and resuspended in equal volumes. AVPL at a final concentration of 30 μg/mL was added to the bacterial suspension, and the sample was incubated at 37 °C for 30 min, followed by recorded spectrophotometer readings. Sterile phosphate buffer was added to the bacterial suspension as a negative control. The experiments were performed in triplicate.

To assess the time and concentration dependence of the effect of lysin AVPL against different strains, cell suspensions of *S. suis* strains SC19 (SS2) and SC493 (SS9) were treated with different concentrations (0, 50, 100, 200, and 300 μg/mL) of AVPL and incubated at 37 °C for 1 h. Samples were taken at 10 min intervals and serially diluted in PBS solution to determine the number of viable bacteria. The cell suspension was treated with sterile phosphate buffer as a negative control. The experiments were performed in triplicate.

### 4.4. Determination of the Biofilm-Forming Ability of S. suis

The 96-well microplate method for determining the ability of *S. suis* to form biofilms was used according to a previous method with some modifications [49]. Briefly, *S. suis* cultured at 37 °C for 12 h was diluted in fresh BHI broth at a ratio of 1:100, and 100 μL (1 × 10^6^ CFU) of the dilution was added to a 96-well microplate with 100 μL of BHI broth (Becton, Dickinson and Company, Franklin Lakes, NJ, USA). Sterile BHI broth was used as a negative control. Each group included three samples. Cultures were removed after incubation at 37 °C for 48 h and then washed three times with sterile PBS. After fixation with methanol for 30 min, each well was stained with 1% (*w*/*v*) crystal violet (CV) for 30 min, washed to remove excess dye, and dried at room temperature. Then, 200 μL of 33% glacial acetic acid solution was added to each well for 30 min at 37 °C to dissolve the crystal violet, and the OD_590_ was measured using a Synergy 2 multimode microplate reader (BioTek, Winooski, VT, USA). The biofilm-forming ability of the isolates was classified using the following four classes: 4 × ODc ≤ OD indicated strong, 2 × ODc < OD ≤ 4 × ODc indicated moderate, ODc < OD ≤ 2 × ODc indicated weak, and OD ≤ ODc indicated non-biofilm forming. ODc was the cutoff value of the OD measurement of the negative control (BHI group) [57]. The experiments were performed in triplicate.

### 4.5. The Activity of AVPL against S. suis Biofilms

SS9 strain SC493 was used to examine AVPL-mediated prevention of biofilms. Briefly, SC493 incubated at 37 °C for 12 h was diluted with sterile BHI broth and transferred to a 96-well microtiter plate at 100 μL per well (1 × 10^6^ CFU). To determine the preventive effect, 100 μL of BHI broth containing a final concentration of 20 μg/mL AVPL was added to each well, and sterile BHI broth (100 μL) was added as a positive control under the same conditions. Further, 200 μL of sterile BHI broth per well served as a negative control. Each group included three samples. After culturing at 37 °C for 48 h or 72 h without shaking, the residual biofilms were stained with CV to determine the OD_590_.

The effect of AVPL on biofilm removal was also examined. After culturing SC493 for 48 h or 72 h to form a biofilm, as described above, the wells were rinsed three times with sterile phosphate buffer. Then, 200 μL of AVPL (final concentration of 20 μg/mL) was added, and the samples were incubated at 37 °C for 1 h; sterile phosphate buffer was added as a negative control. Each group included three samples. After incubation, residual biofilms were stained with CV, and the OD_590_ was determined.

### 4.6. Observation of Biofilms Using Scanning Electron Microscopy (SEM)

The biofilm samples for SEM observation were prepared according to a previous procedure with some modifications [49]. A total of 1 × 10^6^ CFU of *S. suis* SC493 was added to each well of a 12-well microtiter plate with or without 20 μg/mL AVPL in the wells to determine the inhibition of biofilm formation; sterile glass coupons were used to provide an effective site for biofilm adhesion. The 12-well microtiter plates were then incubated at 37 °C for 48 h without shaking, and the wells were washed three times with sterile PBS after removing the cultures. The elimination of biofilm was determined by treatment with or without 20 μg/mL of AVPL at 37 °C for 1 h. Afterward, the biofilms were fixed with 4% glutaraldehyde for 2 h and dried after dehydration with different concentrations of ethanol (10%, 20%, 50%, 70%, 90%, and 100%). Samples were coated with gold, and observation was completed using an SEM.

### 4.7. Binding Activity of Different Domains of AVPL

To investigate the binding of different structural domains of AVPL to *S. suis*, SC19 cells cultured to OD_600_ = 0.8 were incubated with Hoechst 33342 fluorescent dye for 10 min at 37 °C. After washing 5 times with PBS buffer to remove the unbound dye and resuspending in 100 μL, bacterial suspensions were incubated with different domains fused with an enhanced green fluorescent protein (EGFP-AVPL-C, EGFP-AVPL-B1, EGFP-AVPL-B2, or EGFP-AVPL-B1B2) for 20 min at 37 °C; then, cells were collected via centrifugation (8000× *g*, 5 min). After washing to remove unbound proteins, the cells were suspended in 100 μL of PBS buffer, and the fluorescence of the cells was determined via laser scanning confocal microscopy (LSCM) at different excitation wavelengths.

### 4.8. The Therapeutic Effect of AVPL In Vivo

After a period of acclimatization, different amounts of multidrug-resistant SS2 strain SC19 (7 × 10^7^, 7 × 10^8^, or 7 × 10^9^ CFU/mouse) were intraperitoneally injected into each group of mice (*n* = 10) to determine the minimum lethal dose (MLD), with 100% mortality within 7 days.

To determine the protective dose of AVPL for mice infected with a lethal dose of *S. suis*, mice were challenged with a bacterial level that caused 100% mortality within 24 h. After 1 h of infection, AVPL (50, 150, or 300 μg/mouse, *n* = 10) was injected in the same manner. The control group was given equal amounts of phosphate buffer. The survival rate in each group was recorded within the 7-day follow-up period.

In a series of experiments conducted to evaluate the effect of a single intraperitoneal injection of AVPL, mice (*n* = 25 per group) were given AVPL (300 μg/mouse) or phosphate buffer 1 h after SC19 challenge, and the control group was given sterile phosphate buffer. Peripheral blood samples were collected from the tail vein of 3 randomly selected mice in each group at 1–10 h (measured every 1 h interval), 12 h, 24 h, and 48 h postinfection. Three mice were randomly euthanized in each group at 1 h, 6 h, 12 h, 24 h, and 48 h postinfection, and the aseptically collected lung, liver, spleen, and kidney were weighed and homogenized in 1 mL of PBS. Bacterial load in the blood and organs was determined using BHI agar plates. Serum levels of interleukin 6 (IL-6) and tumor necrosis factor-α (TNF-α) were measured using enzyme-linked immunosorbent assay kits (BioLegend, San Diego, CA, USA) according to the manufacturer’s instructions. Three mice in each group were randomly euthanized 24 h and 48 h post-challenge for gross autopsy. Lung, liver, spleen, and kidney tissues were collected and fixed in 4% formalin. Paraffin embedding, sectioning, and hematoxylin–eosin (H&E) staining were followed by light microscopy observation. Serum and tissues from healthy mice were used as controls. To determine bacterial resistance to AVPL in vivo, a total of 90 single colonies from the blood and each organ were collected 48 h after AVPL treatment to determine susceptibility to the lytic enzyme AVPL.

Mice (*n* = 5) were injected intraperitoneally with 600 µg/mouse AVPL or phosphate buffer for analysis of AVPL toxicity. The appearance and behavior of mice were observed for 7 consecutive days for health scoring (0–5) [58]. After 7 days, the mice were euthanized, and the organs were collected and fixed in 4% formalin. The fixed tissues were stained with H&E for microscopic observation.

### 4.9. Statistical Analysis

All statistical analyses were performed using SPSS version 13.0 software (SPSS, Inc., Chicago, IL, USA). The significance of the experimental data was determined with a nonparametric Kruskal–Wallis test followed by pairwise comparisons. * *p* < 0.05, ** *p* < 0.01, and *** *p* < 0.001 were considered to indicate significance. GraphPad Prism 9 (GraphPad Software, Inc., San Diego, CA, USA) was used for chart generation. The error bars represent the standard deviation (SD) of the mean.

## 5. Conclusions

In this study, we report that AVPL, a native phage lysin with broad catalytic activity, shows efficient bactericidal activity against *S. suis*. AVPL can lyse multiple serotypes of *S. suis,* including serotypes 2, 3, 7, 9, 12, and 27, and effectively prevent biofilm formation and eliminate *S. suis* biofilms. A single treatment with AVPL significantly reduced the bacterial load in infected mice, attenuated pathological damage to tissues, and efficiently protected mice with systemic *S. suis* infection. Thus, the *A. viridans* phage lysin AVPL may be a promising therapeutic nonantibiotic agent in the control of multidrug-resistant *S. suis* infections.

## Figures and Tables

**Figure 1 ijms-24-16670-f001:**
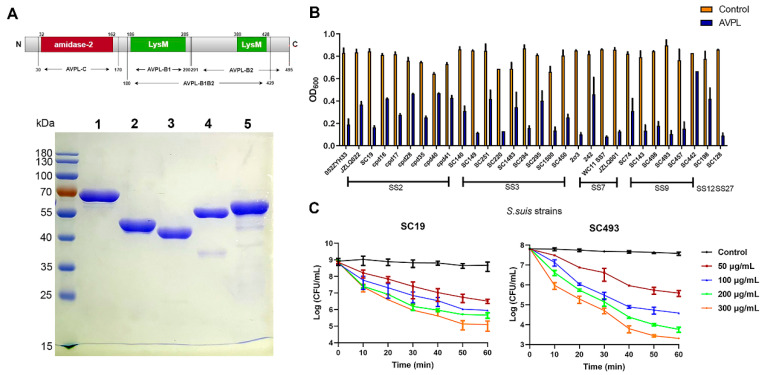
Lytic spectrum and bactericidal activity of AVPL against *S. suis*. (**A**) Schematic representation of AVPL and SDS-page analyses of the recombinant proteins. The lanes were loaded as follows: lane 1, purified AVPL; lane 2, purified EGFP-AVPL-C; lane 3, purified EGFP-AVPL-B1; lane 4, purified EGFP-AVPL-B2; and lane 5, purified EGFP-AVPL-B1B2. (**B**) Susceptibility of *S. suis* to AVPL. Each strain was treated with 30 μg/mL AVPL or phosphate buffer for 30 min, and the decrease in the OD_600_ was recorded by a microplate reader. SS2, SS3, SS7, SS9, SS12, and SS27 represent serotypes of *S. suis*. (**C**) Bactericidal activity of AVPL against *S. suis* strains. Strains were treated with AVPL (0, 50, 100, 200, or 300 μg/mL) or phosphate buffer for 60 min before dilution and plating on BHI agar. The data are shown as the mean and standard deviation (SD) (*n* = 3).

**Figure 2 ijms-24-16670-f002:**
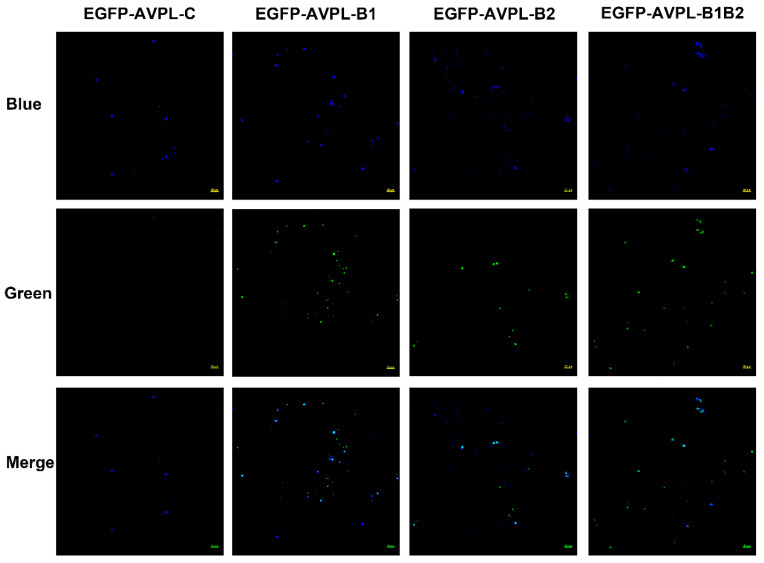
Binding activity of truncated fragments of AVPL to *S. suis*. Strains were stained with 20 µM Hoechst No. 33342 at 37 °C for 10 min and incubated with EGFP-AVPL-C, EGFP-AVPL-B1, EGFP-AVPL-B2, or EGFP-AVPL-B1B2 for 20 min, followed by fluorescence microscopy to visualize the bacteria. Blue: localization at 405 nm and emitted by Hoechst No. 33342; green: localization at 488 nm and emitted by EGFP. The merging of blue and green indicates the colocalization of the recombinant protein with *S. suis*. Scale bar, 20 µm.

**Figure 3 ijms-24-16670-f003:**
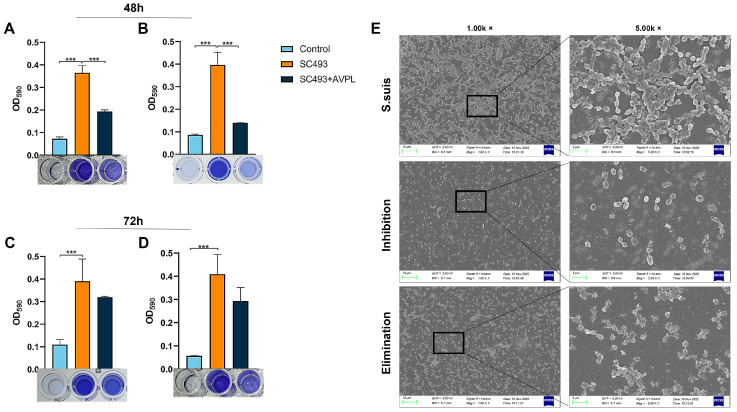
Antibiofilm activity of AVPL. (**A**,**B**) Crystal violet staining for the preventive and clearing effects of AVPL on 48-h biofilms. (**C**,**D**) Crystal violet staining for the preventive and clearing effects of AVPL on 72-h biofilms. Data are shown as the mean ± SD of triplicate experiments. *** *p* < 0.001. (**E**) SEM images of *S. suis* biofilms. The images show the biofilm formed by *S. suis* incubation for 48 h and the biofilm morphology with AVPL pre/posttreatment. The scale bars in the left images indicate 10 µm, and those in the right images indicate 2 µm.

**Figure 4 ijms-24-16670-f004:**
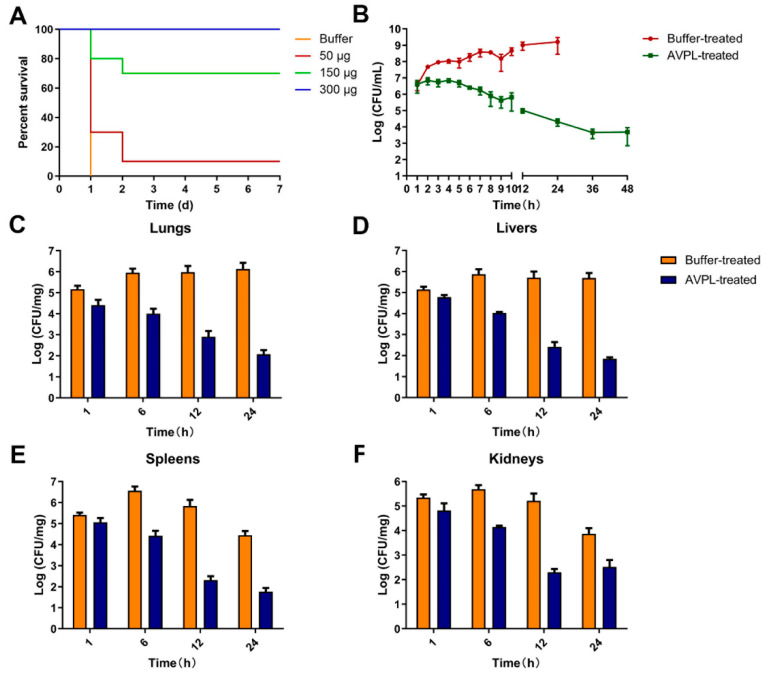
AVPL protects mice from lethal *S. suis* infection. (**A**) Protective efficacy of AVPL. Mice challenged with 7 × 10^8^ CFU/mouse SC19 were randomized into 4 groups. Each group (*n* = 10) intraperitoneally received a single dose of sterile phosphate buffer or different doses of AVPL at 1 h postinfection. (**B**) The bacterial burden in blood was determined. Peripheral blood of mice was obtained at the indicated time points to determine the bacterial load. The bacterial burden in lungs (**C**), livers (**D**), spleens (**E**), and kidneys (**F**) was determined. Mice were euthanized at 1 h, 6 h, 12 h, and 24 h postinfection, and the organs of the different treatment groups were collected aseptically and homogenized to determine the bacterial load (*n* = 3). The above values represent the mean ± SD (*n* = 3).

**Figure 5 ijms-24-16670-f005:**
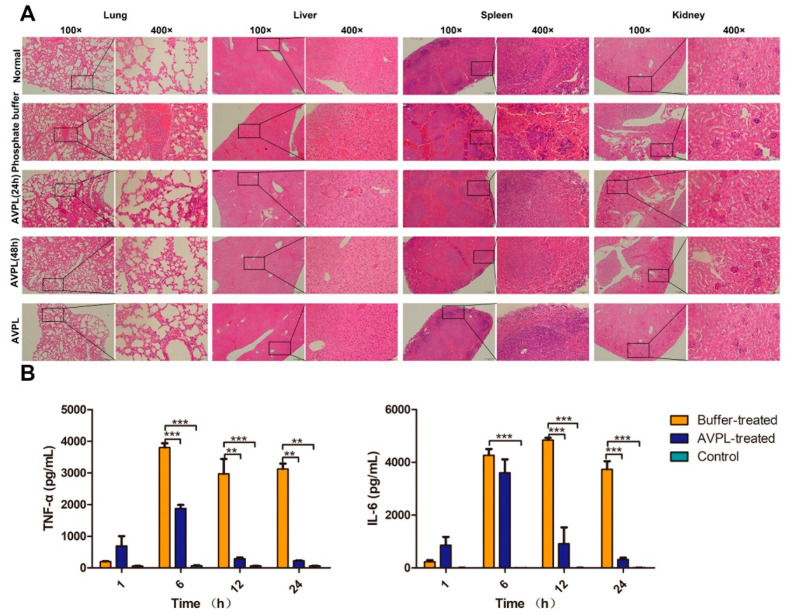
Pathological changes and cytokine levels. (**A**) Histopathology following *S. suis* infection. Lungs, liver, spleen, and kidneys from each group of mice were taken and stained with H&E (magnification, ×100, ×400) at 24 h and 48 h postinfection. A single dose of 600 µg/mouse AVPL was administered intraperitoneally for safety analysis. (**B**) The cytokine levels in mice were determined. IL-6 and TNF-α serum levels were measured in each group of mice at 1 h, 6 h, 12 h, and 24 h postinfection. Sera from healthy mice were used as controls. ** *p* < 0.01, *** *p* < 0.001. Data represent the mean ± SD (*n* = 3).

## Data Availability

The datasets presented in this study can be found in online repositories. The names of the repository/repositories and accession number(s) can be found in the article.

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
