# Peer review of "Aerococcus viridans Phage Lysin AVPL Had Lytic Activity against Streptococcus suis in a Mouse Bacteremia Model"

_ijms, 2023, doi:10.3390/ijms242316670_

Round 1

Reviewer 1 Report

Comments and Suggestions for Authors

This manuscript reports on the use of a lysin named AVPL, isolated from Aerococcus viridans phage, to lyse Streptococcus suis in culture and in a mouse model of infection. The topic is interesting but the manuscript falls short of basic data about the source, molecular cloning and catalytic characteristics of AVPL, and about the preparation of the protein constructs used in the work. All this essential information is dispatched by the authors with a simple mention (not any reference) to their previous unpublished studies. This is not acceptable.

MAJOR ISSUES

1.- In lines 53-54, the authors state “Our previous studies have demonstrated that AVPL is a lysin derived from the A. viridans phage vB_AviM_AVP (GenBank accession: MH729379) (unpubl. data).” The demonstration that AVPL is a lysin (a peptidoglycan hydrolase derived from phages), is essential information that is unpublished. This information should be disclosed and, if there is no reference supporting it, the data should be included in the manuscript.

2.- The GenBank accession for the full sequence of the A. viridans phage vB_AviM_AVP is provided. However, GenBank accessions for the coding sequence and the protein sequence of AVPL are not provided. They are needed.

3.- In lines 61-62, the authors state “The following five recombinant proteins were obtained according to a previously described method, as shown in Figure 1A.”. However, the figure does not show any method to obtain the recombinant proteins. In section 4.3, it is stated that the recombinant proteins “were constructed and preserved as described in previous studies (unpubl. data). Proteins were expressed, purified, and prepared according to the previous description.” Again, all of this is essential information that is unpublished. This information should be fully disclosed and, if there is no reference supporting it, the data should be included in the manuscript.

4.- Besides describing the procedures to produce the recombinant proteins, a scheme summarizing their structures with detail should be added to the manuscript.

5.- The nomenclature used for the recombinant fusions of EGFP and AVPL should make clear to what part of fusion proteins the domains correspond: B1, B2, C are  AVPL domains, therefore never leave the domain name between AVPL and EGFP. For instance, the name EGFP-AVPL-C makes clear that the C domain pertains to AVPL, but this not so in the other cases. I suggest to use EGFP-AVPL-B1, EGFP-AVPL-B2 and EGFP-AVPL-B1B2.

6.- Figure 2. How do you explain that blue and green do not colocalize always? In the “merge” images, blue and green dots can be seen.

MINOR ISSUES

Organism names should be in italics, starting with the title of the manuscript and elsewhere..

Abbreviations should be defined at their first use in the abstract, main text or illustrations.

In Figure 1B, S. suis serotypes 2, 3, 7, 9, 12, and 27 are indicated but are not identified as serotypes. This should be explicit.

Line 68. 15/24 shouldn’t be 15/30?

Line 99. Change “µm/L” to either “µmol/L” or just “µM”.

Line 127. Change “error” to “scale”.

Institutional Review Board Statement for animal research is missing

Many references are incomplete: 1, 3, 6, 7, 11, 12, 13, 18, 21, 24, 28, 29, 31, 34, 41, 43, 46, 47, 48, 49, 50 and 55.

In the footer of Table S1 it is stated that “The numbers represent the serotypes of the different strains of S. suis, respectively”. However, these numbers are not shown in the table.

In Table S2, meropenem is abbreviated as MEM in the head, and ad MEN in the footer.

Comments on the Quality of English Language

Minor edits are required

Reviewer 2 Report

Comments and Suggestions for Authors

This study describes the effect of phage lysin on the inhibition of Streptococcus in vitro and in animal model. There are some doubts and suggestions;

1.     Since phage lysin is considered as antibiotic alternative, the experiments should be conducted with antibiotic-sensitive and antibiotic-resistant Streptococcus suis.

2.     Lysin is peptidoglycan hydrolase. Then, lysin itself is not much effective against biofilm cells that are covered by EPS. It is combined with phage depolymerase should be better than alone. Otherwise, the biofilms used in this study are not true biofilm cells. The antibiofilm mechanism of lysin should be discussed in detail.

3.     The same concentrations used in in vitro study were used for animal study. Does it make sense pharmacokinetically?

4.     Cell culture study, for example macrophage cell line, should be preceded.

Comments on the Quality of English Language

 Moderate editing of English language required

Round 2

Reviewer 1 Report

Comments and Suggestions for Authors

The manuscript has been improved but several points remain to be solved.

MAJOR ISSUE

1.- In the revised version (line 55) the assertion that AVPL is a lysin is based just in GenBank record AXY81212 submitted by some of the authors, but there is no experimental evidence showing that AVPL is a peptidoglycan hydrolase. Of course, the authors demonstrate now that AVPL has lytic activity on S. suis. If this is considered enough to identify AVPL as a lysin, then the identification as such is carried out in the manuscript, not earlier. As a minimum, the authors should explain in what evidence was based their initial hypothesis that GenBank record AXY81212 corresponded to a lysin.

MINOR ISSUES

2.- In line 63, the authors still state “The following five recombinant proteins were obtained according to a previously described method, as shown in Figure 1A.”. However, I must insist, the figure contains an SDS-PAGE gel where the recombinant proteins are shown, not the method to obtain them. Actually, the method to obtain the proteins is now described in Section 4.2.

Line 131. Change “error” to “scale”. (This was already indicated in my first report but the authors failed to correct it).

In the footer of Table S1, it is stated that “The numbers represent the serotypes of the different strains of S. suis, respectively”. In their cover letter, the authors state that these are the numbers that appear as superscripts of strain names. I suggest to change the footer to read “The superscript numbers…

Comments on the Quality of English Language

Minor edits required

Reviewer 2 Report

Comments and Suggestions for Authors

It has been well revised.

Comments on the Quality of English Language

Moderate editing of English language required

Author Response

Response to Comments on the Quality of English Language:

Response: Thanks for your comment. The language of the paper has been further modified.